# Genomic Engineering of Oral Keratinocytes to Establish In Vitro Oral Potentially Malignant Disease Models as a Platform for Treatment Investigation

**DOI:** 10.3390/cells13080710

**Published:** 2024-04-19

**Authors:** Leon J. Wils, Marijke Buijze, Marijke Stigter-van Walsum, Arjen Brink, Britt E. van Kempen, Laura Peferoen, Elisabeth R. Brouns, Jan G. A. M. de Visscher, Erik H. van der Meij, Elisabeth Bloemena, Jos B. Poell, Ruud H. Brakenhoff

**Affiliations:** 1Amsterdam UMC Location Vrije Universiteit Amsterdam, Oral and Maxillofacial Surgery and Oral Pathology, 1081 HV Amsterdam, The Netherlands; l.wils@amsterdamumc.nl (L.J.W.); jgamdevisscher@gmail.com (J.G.A.M.d.V.); e.bloemena@amsterdamumc.nl (E.B.); 2Amsterdam UMC Location Vrije Universiteit Amsterdam, Otolaryngology and Head & Neck Surgery, 1081 HV Amsterdam, The Netherlandsa.brink@amsterdamumc.nl (A.B.);; 3Amsterdam UMC Location Vrije Universiteit Amsterdam, Pathology, 1081 HV Amsterdam, The Netherlands; l.peferoen@amsterdamumc.nl; 4Academic Centre for Dentistry Amsterdam (ACTA), 1081 LA Amsterdam, The Netherlands; 5Cancer Center Amsterdam (CCA), Cancer Biology and Immunology, 1081 HV Amsterdam, The Netherlands

**Keywords:** oral diseases, oral leukoplakia, oral squamous cell carcinoma, malignant transformation, CRISPR/Cas9, cell culture models, small molecule inhibitors, genetic engineering

## Abstract

Precancerous cells in the oral cavity may appear as oral potentially malignant disorders, but they may also present as dysplasia without visual manifestation in tumor-adjacent tissue. As it is currently not possible to prevent the malignant transformation of these oral precancers, new treatments are urgently awaited. Here, we generated precancer culture models using a previously established method for the generation of oral keratinocyte cultures and incorporated CRISPR/Cas9 editing. The generated cell lines were used to investigate the efficacy of a set of small molecule inhibitors. Tumor-adjacent mucosa and oral leukoplakia biopsies were cultured and genetically characterized. Mutations were introduced in *CDKN2A* and *TP53* using CRISPR/Cas9 and combined with the ectopic activation of telomerase to generate cell lines with prolonged proliferation. The method was tested in normal oral keratinocytes and tumor-adjacent biopsies and subsequently applied to a large set of oral leukoplakia biopsies. Finally, a subset of the immortalized cell lines was used to assess the efficacy of a set of small molecule inhibitors. Culturing and genomic engineering was highly efficient for normal and tumor-adjacent oral keratinocytes, but success rates in oral leukoplakia were remarkably low. Knock-out of *CDKN2A* in combination with either the activation of telomerase or knock-out of *TP53* seemed a prerequisite for immortalization. Prolonged culturing was accompanied by additional genetic aberrations in these cultures. The generated cell lines were more sensitive than normal keratinocytes to small molecule inhibitors of previously identified targets. In conclusion, while very effective for normal keratinocytes and tumor-adjacent biopsies, the success rate of oral leukoplakia cell culturing methods was very low. Genomic engineering enabled the prolonged culturing of OL-derived keratinocytes but was associated with acquired genetic changes. Further studies are required to assess to what extent the immortalized cultures faithfully represent characteristics of the cells in vivo.

## 1. Introduction

Head and neck squamous cell cancer (HNSCC) is the sixth most common type of cancer, with a worldwide incidence of 880,000 cases [1]. Half of HNSCC tumors arise within the oral cavity as oral squamous cell carcinoma (OSCC) [1]. Most OSCCs seem to arise de novo, but several lines of evidence indicate that these emerge from precancerous mucosal changes that are not macroscopically visible as lesions. Such precancerous changes, often indicated as ‘fields’, are identified by histology and/or genetic markers in tumor-adjacent mucosa and surgical margins [2,3]. In treated oral cancer patients, these fields often remain behind, unnoticed by their lack of visible appearance, causing local recurrences and second primary tumors [2,4].

Some of the precancerous mucosal changes that precede the tumors are visible as lesions, which are collectively known as oral potentially malignant disorders (OPMDs) [5]. The most common OPMD is oral leukoplakia (OL), which is defined by the WHO as “A white plaque of questionable risk, having excluded (other) known diseases or disorders that carry no increased risk for cancer” [6,7]. The reported worldwide prevalence of OL is 2–4% and the annual malignant transformation rate to OSCC is estimated to be 1–5% [6,7,8,9,10]. 

The standard of care for OL consists of anamnesis and clinical inspection as well as the identification and cessation of possible risk factors. Lesions are biopsied to exclude the presence of invasive carcinoma and identify the presence and grade of dysplasia. Small lesions are generally excised, but for patients who present with unresectable or multifocal lesions, only an incisional biopsy is obtained [5,11]. There is currently no evidence that surgical excision, or any other treatment, prevents malignant transformation in these patients, although there is a lack of randomized clinical studies to demonstrate this convincingly [9,12,13,14]. Lesions often recur at the same site in the oral cavity, but OL patients are also at increased risk of developing oral cancer outside the lesion, likely by field cancerization [13,15,16]. Consequently, all OL patients need to remain in frequent clinical follow-up at dedicated oral medicine and cancer centers. 

As surgery is not an effective ablative therapy to prevent the malignant transformation of OL, a wide range of chemopreventive and natural compounds for the treatment of OL have been investigated in the past or are currently under evaluation in clinical trials [9,12]. These interventions target cellular pathways and processes that may play a role in the malignant transformation of OL such as the cell cycle (bleomycin), epithelial cell proliferation and differentiation (vitamin A and epidermal growth factor receptor inhibitors erlotinib, vandetanib, and cetuximab), glucose metabolism (anti-diabetic drugs such as metformin and pioglitazone), inflammation by non-steroidal anti-inflammatory drugs (celecoxib, aspirin, and ketorolac), and reduced immune response by immunotherapy (pembrolizumab, nivolumab, and avelumab). Although some of these drugs showed promising results in the reduction or even removal of OL, there was no correlation with the prevention of OL recurrence or malignant transformation of OL and, in various studies, side effects were reported [9,12,17,18,19,20,21,22,23,24,25,26,27,28,29,30,31,32,33,34,35]. Convincing evidence from randomized controlled trials that oral cancer can be prevented is still lacking at present. 

There is a need for therapies that are minimally invasive and more directed towards OL lesions specifically but that also allow the simultaneous targeting of the surrounding altered fields that are clinically not visible. Pathways upon which precancer cells rely more heavily than their healthy counterparts may present opportunities for novel therapies. Tumor suppressor genes *TP53* and *CDKN2A* are important cell cycle regulators that inhibit cell division in response to a lack of mitogenic stimuli or DNA damage. These genes are often inactivated in OL and OSCCs, which opens the possibility for the exploitation of synthetic lethality [36,37]. Recent genomic screens for HNSCC and oral precancer have identified multiple druggable targets that play a role in cell cycle regulation in these (pre)malignant cells such as *PLK1*, *WEE1*, and *CHEK1* [38,39,40]. A number of first-generation drugs targeting these genes have been assessed in HNSCC and tumor-adjacent oral precancer models, but, in vivo, toxicities might hamper clinical application, especially when considering that OL is not yet malignant [38,39,41,42]. In order to further validate the efficacy of targeting these genes by novel small molecule inhibitors and identify additional targets and treatments, research in representative cell models of OL is of major importance. 

Research in vitro has contributed to our understanding of the biological mechanisms underlying oral carcinogenesis and OL treatment efficacy. Primary oral keratinocytes in cultures increase the expression of p16^INK4a^ (p16), encoded by the *CDKN2A* gene, and undergo cell cycle arrest by premature senescence [43]. Prolonged cell proliferation is also hampered by telomere erosion, the classical form of replicative senescence [43]. This is reflected by the fact that almost all oral precancer cell models established to date have lost the expression of p16 and display increased levels of telomerase activity [43,44,45,46,47]. While these cell lines have been studied extensively, they do not capture all precancerous changes observed in OL patients. The number of head and neck cancer driver genes encompasses over 30 candidates, and how these interact in carcinogenesis is largely unknown. 

We have previously shown that it is possible to generate cultures from normal keratinocytes and OSCC adjacent tissue biopsies, resulting in oral precancer cell lines with an extended although still limited lifespan that may subsequently be used for the investigation of novel preventive treatments [39,40,47,48]. The primary aim of the present study was to assess whether this same technique could be used to culture keratinocytes from OL biopsies, and the secondary aim was to characterize the genetic factors involved. We first show that by the addition of the *TERT* gene, it is possible to immortalize an OSCC tumor-adjacent cell culture. Subsequently, we demonstrate that it is possible to generate immortalized cell lines from normal oral keratinocytes and tumor-adjacent oral biopsies using genomic engineering to modify selected genes. This same approach was applied to generate immortalized OL cell lines, and these cultures were genetically characterized over time. Finally, a subset of immortalized cell lines was used to assess the response of a set of targeted small molecule inhibitors. 

## 2. Materials and Methods

### 2.1. Patient Material

For this study, biopsies were obtained from patients during routine visits at the Department of Oral and Maxillofacial Surgery/Oral Pathology at Amsterdam UMC, location VUmc, The Netherlands, between 1 January 2019 and 31 August 2022. OL biopsies were either obtained at the time of initial OL diagnosis or during a follow-up visit. Of each biopsy, one-half was used for routine histopathological examination to confirm the clinical OL diagnosis and exclude the presence of invasive OSCC. The other half was kept at 4 °C in Keratinocyte Growth Medium (KGM, KBM Gold Basal Medium, cat. no. 00192151, Lonza, Verviers, Belgium) supplemented with 0.1% bovine serum albumin (BSA, A7159, Sigma Aldrich, Zwijndrecht, The Netherlands) and KGM Gold SingleQuots supplements (containing insulin, hydrocortisone, human recombinant EGF, transferrin, epinephrin, GA-1000, and Bovine Pituitary Extract, cat. no. 00192060, Lonza, Verviers, Belgium) until it was processed as described below. Tumor-adjacent tissue biopsies were obtained from patients after tumor resection. These tumor-adjacent biopsies were processed similarly to the OL biopsies. 

The current study followed the principles of the Helsinki Declaration and the national guidelines for the secondary use of human tissue of the Dutch Federation of Biomedical Scientific Societies and the General Data Protection Regulation of the European Commission. This study was performed in accordance with the guidelines of the Institutional Research Board of Amsterdam UMC (FWA00017598). This committee confirmed that the Dutch Medical Research Involving Human Subjects Act (WMO) did not apply to this study and approved the present study under 2014.139 (oral leukoplakia biopsies), 2015.345 (UPPP), and 2016.035 (tumor-adjacent tissue biopsies). Written informed consent was obtained from all patients or waived (UPPP). 

### 2.2. Processing of Oral Leukoplakia and Tumor-Adjacent Biopsies

Surgically removed fresh OL and tumor-adjacent biopsies were cut into smaller pieces (depending on biopsy size) and transferred to a 1X Dispase II (cat. no. 4942078001, Roche, Basel, Switzerland) and phosphate-buffered saline (PBS, cat. no. BE17-517Q, Lonza, Verviers, Belgium) solution and incubated at room temperature (RT) for 4 h to allow diffusion. Subsequently, the vial was incubated at 37 °C for 15 min to activate the dispase and allow for the separation of the mucosal epithelium and submucosal layers. Whenever possible, the separated mucosal layer was transferred to a vial containing preheated TrypLE Express (cat. no. 12605036, Thermo Fisher Scientific, Bleiswijk, The Netherlands) at 37 °C for 10 min. Dissociated cells were rinsed with KGM once and plated in a 12-well plate for further culturing. The remaining tissue or complete biopsies when tissue layers could not be separated were also treated by TrypLE express, rinsed, and cultured in KGM. 

### 2.3. Cell Culturing

Cells were cultured at 37 °C in 5% CO_2_ (Heracell 150, Thermo Fisher Scientific, Bleiswijk, The Netherlands). All cells were cultured in KGM supplemented with growth factors and 1% BSA. In addition to the primary oral keratinocytes obtained from fresh patient material, several already-established precancer and tumor cell lines were cultured and used as controls in multiple experiments. Precancer cell line VU-preSCC-M3 (M3) [48], tumor cell line VU-SCC-040 (040) [49], and normal oral keratinocyte cell line VU-UPPP60 [47] were previously generated in our own department. Tumor cell lines UM-SCC-11B (11B) [50], UM-SCC-17A (17A) [50], UM-SCC-22A (22A) [50], and UM-SCC-47 (47) [50] were obtained from Prof T. Carey (University of Michigan, Ann Arbor, MI, USA). Human breast cancer cell line MCF-7 [51], used as a control in several experiments, and the HEK293T cell line, used for lentiviral packaging, were both acquired from ATCC (MCF7: HTB-22; HEK293T: 293tsA1609neo; ATCC, Manasses, VI, USA). HEK293T and all tumor cell lines were cultured in Dulbecco’s Modified Eagle’s Medium (DMEM, cat. no. BE15-604K, Lonza, Verviers, Belgium) supplemented with 5% fetal bovine serum (FBS, 04-007-1 A, Biological Industries, Kibbutz Beit-Haemek, Israel) and 2 mmol/L L-glutamine (BE17-605F, Lonza, Verviers, Belgium). All cell lines were passaged when they reached 60% confluency or more. TrypLE Express was used to detach cells, after which, a medium was added and the suspended cells were transferred to a new vial. Subsequently, tumor cells were plated in a new flask directly. PreSCC and normal keratinocyte suspensions were first spun down at 300× *g* for 5 min, after which, the supernatant was removed, the cell pellet was resuspended in KGM-BSA, and the cells were plated, generally in 1:5 dilutions. Primary cell cultures from fresh biopsies were defined as ‘proliferating’ when the cells grew to >60% confluency and could be passaged at least once. Population doublings were calculated at every passage from the moment cells started to proliferate, usually oligoclonal proliferation. Cell cultures were considered to have an extended lifespan after 25 population doublings [47]. Cell cultures were defined as immortal when the cells reached 50 population doublings without showing progressive signs of senescence. All cell cultures were regularly checked for the presence of mycoplasma (MycoAlert, LT07-318, Lonza, Verviers, Belgium) and authenticated by visual inspection and DNA sequencing on indication.

### 2.4. DNA Library Preparation

Genomic DNA was isolated from cell pellets using the PureLink genomic DNA mini kit (cat. no. 2485222, Invitrogen, Waltham, MA, USA) and subsequently fragmented using Covaris sonication (ME220, Covaris, Inc., Woburn, MA, USA) with an aimed average length between 180 and 220 base pairs. Fragmented DNA was cleaned up using KAPA HyperPure beads at 1.6× (cat. no. 08963851001, Roche, Basel, Switzerland). The concentration of the fragmented DNA was determined using the Qubit dsDNA High Sensitivity kit (cat. no. Q32854, Thermo Fisher Scientific, Bleiswijk, The Netherlands). Per sample, 100 ng of DNA was used for DNA library preparation using the KAPA HyperPrep Kit (cat. no. 07962363001, Roche, Basel, Switzerland) according to the protocol of the vendor (KAPA HyperCap workflow v3.0). End repair and A-tailing were performed, followed by ligation of the KAPA universal adapter and a 0.8× clean-up. The adapter-ligated samples were amplified in the presence of KAPA UDI primer mixes by an 8-cycle PCR followed by a 1× clean-up and final concentration determination using the Bioanalyzer DNA high-sensitivity chip (2100 Bioanalyzer system, Agilent, Santa Clara, CA, USA). Samples were pooled equimolarly to a final concentration of 10 nM. This pool was used for 150 bp paired-end sequencing on the NovaSeq 6000 platform (Illumina, San Diego, CA, USA) to obtain low-coverage, whole-genome profiles for the analysis of copy number aberrations (CNA). 

The same DNA libraries were used for the target enrichment analysis of 30 genes often mutated in HNSCC: AJUBA, ASXL1, B2M, CASP8, CDKN2A, CUL3, DDX3X, EPHA2, FAT1, FBXW7, FGFR3, HRAS, KDM6A, KEAP1, KMT2D, KRAS, NFE2L2, NOTCH1, NSD1, OR4A5, PIK3CA, PTEN, RAC1, RB1, RHOA, TERT (promotor region only), TGFBR2, TP53 (including the 5′-UTR), TP63, and ZNF750. The capture was performed using the KAPA HyperCap kit (cat. no. 9075810001, Roche, Basel, Switzerland) according to the protocol of the vendor (KAPA HyperCap workflow v3.2). DNA libraries were thawed on ice and 120 ng of DNA from each sample was pooled to at least 2000 ng per pool (approximately 17–24 samples). Pools were mixed with COT human DNA and KAPA HyperPure beads were added at a 2X ratio. The bead-bound DNA mixture was resuspended in Universal Enhancing Oligos and mixed with a hybridization buffer and component H. The custom KAPA Target Enrichment Probes were added and incubated at 55 °C for 16–20 h. The hybridized DNA was captured using KAPA HyperCap Capture Beads (cat. no. 09075780001, Roche, Basel, Switzerland) and the DNA–bead mixture was washed to remove unbound DNA. Bound DNA was eluted and amplified in a 16-cycle PCR, which was followed by a 1.4× bead clean-up using KAPA HyperPure beads (cat. no. 09075780001, Roche, Basel, Switzerland). The DNA concentration of the pools was measured using the Bioanalyzer DNA high-sensitivity chip. The final pools were diluted to a concentration of 10 nM and used for 150 bp paired-end sequencing on the NovaSeq 6000 platform.

### 2.5. Somatic Copy Number Calling

Fastq files were aligned to the hg19 genome using bwa mem (bwa 0.7.17) [52]. Overlap in paired reads was clipped using ClipBam (fgbio 2.13, Fulcrum Genomics). Copy number analysis was performed using the QDNAseq R package (QDNAseq 1.34.0) [53] based on read counts in a total of 6206 fixed 500 kb bins. Samples with less than 200,000 reads were excluded from further analysis. Samples were normalized through dewaveBins [54] (QDNAseq), an adaptation of NOWAVES that is amenable to next-generation sequencing (NGS) data, using a separate control data set consisting of 29 normal fresh oral mucosa samples that did not contain any CNAs, obtained from a cohort of healthy elderly individuals for the Longitudinal Aging Study Amsterdam, followed by segmentation [3,55]. This same control data set was used for the calling of CNAs. ACE (ACE 1.16.0) [56] was used to align the segments of the control data set with the samples and CNAs were called as described before [3,36]. 

*CDKN2A* is often lost in OSCC due to small focal deletions in chromosome locus 9p21, which may be missed through conventional CNA analysis with larger bin sizes. To assess the presence of losses of 9p21, an additional copy number analysis was performed using 10 kb bins, using the same methods as described above. Losses and double losses were called for the specific bins that contained the *CDKN2A* gene using ACEcall (ACE 1.16.0) [56]. 

### 2.6. Somatic Mutation Calling

Somatic mutation calling was performed using fastq files which were aligned to the hg19 genomic assembly using bwa mem. Duplicates were removed using MarkDuplicates (Picard, 2.20.1, RRID:SCR_006525, GATK toolkit, 4.3.0.0, RRID:SCR_001876); overlap in paired reads was clipped using ClipBam and metrics were collected (GATK4 toolkit). Mutations were called using two callers: Mutect2 (GATK toolkit) [57] and Varscan2 (Varscan 2, 2.4.4) [58], both using manual settings for a minimal base quality of 20 and lenient variant allele frequency, and annotated using Funcotator (Funcotator, 1.7, GATK toolkit). For each sample, the per-base coverage, including deletions, was calculated, with a minimal quality score of 20. Samples with a median coverage of less than 10 were excluded from further analysis. Mutations were only reported when called by both Mutect2 and Varscan2. Funcotator was used to exclude germline mutations (dbSNP build 142) [59], synonymous mutations, mutations in homopolymers, and likely slippage artifacts (3 base deletions in repeats). The minimum required variant-supporting read depth for calling mutations or deletions was determined for each base of each sample on the basis of an expected maximum error rate of 1 in 100 for the Illumina sequencing platform at a read quality of 20 [60]. *p*-values were calculated using the binomial distribution as the probability of sequencing equally or more variant-supporting reads, given the expected maximum error rate and total depth at the specific base location. Variants were reported with *p*-values < 10^−6^. Insertions were called when VAFs exceeded 10% with a minimum of 10 variant-supporting reads. In addition, we performed germline SNP controls to check the identity of all cell cultures. An overview of all included variants is presented in Appendix A.

### 2.7. CDKN2A Methylation Assay

The EZ DNA Methylation-Gold kit (cat. no. D5005, ZymoResearch, Freiburg im Breisgau, Germany) was used to perform sodium bisulphite conversion of 50 ng of genomic DNA per sample (isolated as described above) according to the protocol of the vendor for a final concentration of 4 ng/µL. Subsequently, a quantitative methylation-specific PCR (qMSP) assay was performed using the EpiTect methylight PCR kit (cat. no. 59496, Qiagen, Venlo, The Netherlands). Per sample, a mix was prepared containing TaqMan Master Mix (10 µL, cat. no. 4324018, Applied Biosystems, Wageningen, The Netherlands), *CDKN2A* forward primer (25 pmol/µL, GCGGTCGTGGTTAGTTAG), *CDKN2A* reverse primer (25 pmol/µL, TACGCTCGACGACTACG), *CDKN2A* probe (25 pmol/µL, 5′-6FAM-AACCGACGACGAAAAACAAC-MGB-3′), *ACTB* forward primer (25 pmol/µL, TGGTGATGGAGGAGGTTTAGTAAGT), *ACTB* reverse primer (25 pmol/µL, AACCAATAAAACCTACTCCTCCCTTAA), *ACTB* probe (25 pmol/µL, 5′-Cy5-ACCACCACCCAACACACAATAACAAACACA-3′), nuclease-free water, and 10 ng of bisulfite-treated DNA for a total of 20 µL per sample. Then, 2X EpiTect master mix and 30 µM Carboxy-X-Rhodamine (CXR) reference dye (C5411, Promega, Leiden, The Netherlands) were added to each well, followed by a 45-cycle PCR using the ABI 7500 Real-Time PCR system (Applied Biosystems, Wageningen, The Netherlands). Tumor line UM-SCC-17A was used as a positive control and to create a standard curve as it has known *CDKN2A* methylation [61]. The *CDKN2A* promotor was defined as methylated at a CT value above the detection threshold of at least more than 1/1000 of the value for UM-SCC-17A. 

### 2.8. Genomic Engineering 

As primary keratinocyte cultures have a very limited proliferation capacity of a few passages, the Edit-R all-in-one lentiviral CRISPR/Cas9 system (all vectors with hEF1α promotor and puromycin selection resistance marker, Horizon Discovery, Cambridge, United Kingdom) was used to create knock-outs (KOs) for selected OSCC genes *CDKN2A* (GSGH11935-247687954) and *TP53* (GSGH11935-247754810), a positive control (*DNMT3B*, GSGH12134), and a non-targeting control (GSGC11963). First, an axenic culture was made for each plasmid on a Luria–Bertoni (LB) plate from the obtained glycerol stocks. Subsequently, a single clone was picked and grown at 37 °C overnight in LB broth. Both the LB plate and LB broth contained ampicillin. The bacterial culture was pelleted and DNA plasmids were isolated using the ZymoPure plasmid midiprep kit (cat. no. D4200, ZymoResearch, Freiburg im Breisgau, Germany). In addition, a plasmid for the expression of telomerase reverse transcriptase (*TERT*, pCDH-TERT, #51631, Addgene, Watertown, MA, USA) was produced likewise. This plasmid contained a neomycin selection resistance marker. 

The packaging of lentivirus particles was performed using HEK293T cells. For each plasmid, a master mix was prepared containing 3125 ng of plasmid DNA, 2812.5 ng of a packaging vector (Sigma Aldrich, Zwijndrecht, The Netherlands), 312.5 ng of an envelope vector (Sigma Aldrich, Zwijndrecht, The Netherlands), 500 µL of DMEM (without serum and antibiotics), and 18.75 µL of polyethylenimine (1 mg/mL, cat. no. 23966, Polyscience, Warrington, PA, USA). After 15 min of incubation at RT, this mix was dropwise added to the HEK293T cells at 1:10 (e.g., ~500 uL mix in 4.500 uL DMEM) and incubated at 37 °C and 5% CO_2_ overnight. After 24 h, the medium was refreshed, and 24 and 48 h later, the medium containing the produced virus supernatant was collected, spun down at 250× *g* for 5 min, aliquoted, snap frozen, and stored at −80 °C until further use. This process was repeated 24 h later.

For virus transduction, cells of interest were seeded at various numbers depending on the cell line used. Transduction was performed at 30–40% confluency. Polybrene (1.6 mg/mL, cat. no. 107689-100G, Sigma Aldrich, Zwijndrecht, The Netherlands) was added to the lentivirus-containing medium at a 1:200 ratio. When two guides were combined, these were first mixed at a 1:1 ratio before polybrene was added. For parental cells, a DMEM medium was mixed with polybrene. For cells that are normally cultured with KGM, the virus/polybrene mix (which was DMEM-based) was added to the cells and incubated for 4 h at 37 °C at 5% CO_2_. For cells that are normally cultured with DMEM, this incubation period was extended to overnight for ~16 h. After incubation, the virus medium was removed, followed by a rinse with PBS, after which, the appropriate medium was added. In general, no active selection was performed as we expected cells without modifications to go into senescence naturally. 

Because of the very limited lifespan of the oral keratinocytes, all gene KOs were performed without prior knowledge of the genetic status of the targeted genes. It was, therefore, possible that already mutated or lost genes were targeted by genomic engineering. The *TERT* encoding construct was introduced at the moment the parental cells stopped dividing.

### 2.9. Western Blot

Cell pellets were lysed using RIPA lysis and an extraction buffer (cat. no. 89901, Thermo Fisher Scientific, Bleiswijk, The Netherlands) containing 1× HALT protease and phosphatase cocktail (cat. no. 1861261, Thermo Fisher Scientific, Bleiswijk, The Netherlands). Protein concentration was determined using the Pierce BCA protein assay kit (cat. no. 23227, Thermo Fisher Scientific, Bleiswijk, The Netherlands), followed by the equalization of all lysates. Samples were run on 4–20% Mini-PROTEAN TGX precast protein gels (cat. no. 4561094/4561096, BIO RAD, Hercules, CA, USA) and transferred to an Immobilon-P PVDF Millipore membrane (cat. no. IPVH00010, Sigma Aldrich, Zwijndrecht, The Netherlands). Western blots were performed using the p16 mouse monoclonal antibody (p16 INK4A (JC8), sc-56330, cat. no. C1121, Santa Cruz Biotechnology, Dallas, TX, USA) and p53 mouse monoclonal antibody (p53 Protein (DO-7), cat. no. M7001, Agilent, Santa Clara, CA, USA) and visualized with a red-fluorescent-label goat anti-mouse secondary antibody (LI-COR IRDye 680 RD, 926-68070, cat. no. D11130-05, LI-COR Biosciences, Lincoln, NE, USA) or ECL. β-actin was included as a loading control using the β-actin rabbit monoclonal antibody (13E5, cat. no. 4970, Cell signaling technologies, Denver, MA, USA) and visualized with a green-fluorescent-label goat anti-rabbit secondary antibody (LI-COR IRDye 800CW, 926-32211, cat. no. C30829-02, LI-COR Biosciences). All blots were scanned using the Odyssey infrared imaging system (LI-COR Biosciences) and analyzed with FIJI/ImageJ (ImageJ, 1.53t, RRID:SCR_003070).

### 2.10. Dose–Response Analysis

Cells were seeded in a 96-well plate at an optimized number of cells per well for each cell line in 100 µL of KGM or DMEM (Appendix A). The plates were incubated at 37 °C in 5% CO_2_ for 24 h. Subsequently, the efficacy of four different small molecule inhibitors was assessed by serial dilutions. Serial dilutions for Wee1 inhibitor MK1775 (cat. no. 2373, Biovision, San Francisco, CA, USA), MCL1 inhibitor S63845 (cat. no. HY-100741, MedChemExpress, Monmouth Junction, NJ, USA), or Chk1 inhibitor LY2606368 (cat. no. HY-18174, MedChemExpress, Monmouth Junction, NJ, USA) ranged from 0.8 nM to 100 µM. For PLK1 inhibitor GSK461364 (cat. no. Axon 1688, Axon Medchem, Reston, VA, USA), a serial dilution ranging from 7.6 pM to 1 µM was used. All cells were incubated with the target drug at 37 °C in 5% CO_2_ for 72 h. Cell viability was determined after 72 h of exposure using the CellTiter-Blue assay (cat. no. G8080, Promega, Leiden, The Netherlands). Incubation with CellTiter-Blue was 3 h for all primary oral keratinocytes and preSCC lines and 2 h for all HNSCC tumor cell lines. Cell viability was analyzed using the Glomax Discover Microplate Reader (cat. no. GM3000, Promega, Leiden, The Netherlands) at a green excitation of 520 nm and emission of 580–640 nm with high sensitivity. All drugs were tested three times in triplicate and data were analyzed using GraphPad Prism (GraphPad Prism 9.2.0, RRID: SCR_002798). 

### 2.11. Telomere Length and Telomerase Activity Assays

The Absolute Human Telomere Length Quantification qPCR Assay Kit (cat. no. 8919, ScienceCell, Carlsbad, CA, USA) was used to calculate the average telomere length per chromosome end according to the protocol of the manufacturer. In short, genomic DNA was isolated from cultured cell panels as described above. Subsequently, two qPCR reactions were performed, one with telomere primers and one with single copy reference (SCR) primers as a reference, and a genomic DNA reference sample was included as a positive control and standard for length calculations. Samples without a DNA template were added as negative controls. The resulting CT values, corrected for the input on the basis of the SCR, were calculated relative to the CT values from the reference sample to calculate absolute telomere length, as described in the protocol of the manufacturer. 

The Telomerase Activity Quantification qPCR Assay Kit (cat. no. 8928, ScienceCell) was used to calculate the relative telomerase activity according to the protocol of the manufacturer. In short, cells were counted and 20 µL/million cells of cell lysis buffer was added, supplemented with 0.2 µL HALT protease inhibitor (100×, cat. no. 1861261, Thermo Fisher Scientific) and 0.3 µL β-mercaptoethanol (Thermo Fisher Scientific). The homogenized samples were left at 4 °C for 30 min and then spun down at 15,000× *g* at 4° C for 20 min, and the supernatant was transferred to a new vial. A telomerase reaction was set up with 0.5 µL of sample, 4 µL of 5× telomerase buffer, and 15.5 µL of nuclease-free H_2_O at 37 °C for 3 h, followed by incubation at 85 °C for 10 min to inactivate the reaction. A qPCR reaction with telomere primers was performed for all samples and a reference sample and telomerase activity were calculated relative to the reference extract in the kit, according to the protocol of the manufacturer. 

## 3. Results

### 3.1. Immortalization of Oral Keratinocytes by Genomic Engineering of Selected Target Genes

The previously established tumor-adjacent cell line M3 has an extended lifespan and proliferates for approximately 25 population doublings. We hypothesized that M3 runs into replicative senescence by telomere erosion. Indeed, normal M3 stopped proliferating after ~25 passages, with almost no telomerase activity and shortening telomeres after passage 25 (Figure 1). Inducing ectopic *TERT* expression resulted in an immortalized cell line, M3-TERT^+^. Telomerase was clearly active and telomeres lengthened throughout culturing (Figure 1). In addition to the expression of *TERT*, this cell line contained a double loss of the 9p21 locus containing the *CDKN2A* gene and multiple mutations in *TP53* and *NOTCH1* as inherited from the parental M3 line (Figure 2). Notably, the M3-TERT^+^ line contained many more copy number alterations compared to the parental M3 line. 

Based on these results and previous research [43,44,45,47], different combinations of *CDKN2A* and *TP53* KOs combined with the expression of *TERT* were tested for their potential to immortalize oral keratinocytes. As a first step, this was investigated in oral keratinocytes obtained from a non-oncologic uvula resection, VU-UPPP60 (UPPP60). These cells normally proliferate for a maximum of ten population doublings, which provides enough time for genomic engineering (Figure 2). All combinations of *CDKN2A* (p16) and *TP53* KOs and *TERT* expression were applied, which resulted in one cell line with an extended lifespan (UPPP60-p16^KO^) and four immortalized cell lines (UPPP60-p16^KO^-p53^KO^, UPPP60-p16^KO^-TERT^+^, UPPP60-p53^KO^-TERT^+^, and UPPP60-p16^KO^-p53^KO^-TERT^+^). KOs were confirmed by NGS and analysis of the depth of coverage and IGV inspection when necessary, and loss of protein expression was confirmed by Western blot (Figure 2, Appendix A). All immortalized cell lines at least contained KO or loss of *CDKN2A*, and *TERT* was ectopically expressed in three out of four cell lines. The ectopic expression of *TERT* was only lacking in lines with *TP53* KOs. Surprisingly, *TP53* was not successfully knocked out in the immortalized UPPP60-p53^KO^-TERT^+^ line, suggesting that the KO of *TP53* contributed marginally to immortalization, at least in these in vitro culture models. These results show that it is possible to generate immortalized cell lines from oral keratinocytes using genomic engineering. However, the immortalized derivative lines all showed a different pattern of losses and gains, suggesting that these were all introduced during prolonged culturing. In addition, it should be noted that there was a *TP53* mutation present in UPPP60, albeit at a very low VAF of 2% (Appendix A, page 10, top left figure). Although this sample was obtained from a non-cancer patient, it apparently contained a somatic mutation in *TP53*. The somatic mutation was not present in any of the immortalized lines, again demonstrating the lack of selective advantage that the loss of *TP53* had in this context. 

Next, we applied this approach with a set of tumor-adjacent biopsies. All five biopsies proliferated successfully for 1–5 population doublings: VU-preSCC-HN1009 (HN1009), VU-preSCC-HN1028 (HN1028), VU-preSCC-HN1029 (HN1029), VU-preSCC-HN1031 (HN1031), and VU-preSCC-HN1037 (HN1037, Figure 3). Table 1 contains the clinical background of the included patients. Four out of five initial cell cultures contained mutations in known OSCC genes. While this was insufficient for extended proliferation, the cells could be kept in the culture long enough to perform genomic engineering. Seven cell lines with extended lifespans and nine immortalized cell lines could be generated (Figure 3). In two cultures, genes were targeted that were retrospectively found to be already mutated in the original culture (*TP53* in HN1031 and *CDKN2A* in HN1037, Table 1 and Figure 3). All 16 extended and immortalized cell lines contained mutations or losses of *CDKN2A* (Figure 3, Appendix A). *TP53* KO was present in two out of seven cell lines with extended lifespans and six out of nine immortalized cell lines (Figure 3, Appendix A). *TERT* was expressed in three out of seven cell lines with extended lifespans and six out of nine immortalized cell lines. While *CDKN2A* KO only was sufficient for extended lifespans, all immortalized cell lines additionally contained KO of *TP53*, the expression of *TERT*, or both. Again, the copy number changes of the immortalized cell lines showed various differences within isogenic cell lines, again suggesting that these were induced during prolonged culturing. 

Our data, as well as published data, indicate that normal oral as well as tumor-adjacent keratinocytes can be cultured efficiently for a few passages and pushed into immortalization by the combined modification of *CDKN2A*, *TP53*, and *TERT*. Following this proof of concept in normal and tumor-adjacent oral keratinocytes, these genomic engineering tools were employed to generate OL-derived cell lines. Between 1 November 2018 and 1 January 2023, in total, 94 OL biopsies were obtained from 85 patients with a definitive diagnosis of OL. All biopsies were cultured as described above. Oral keratinocyte outgrowth was observed in 35 out of 94 biopsies (37%). This was a much lower frequency than that for the cultures from normal mucosal epithelium and tumor-adjacent biopsies with success rates of >90%. These 35 cultures proliferated for a median of 2 population doublings (range 1–8). To identify potentially relevant genetic changes associated with culturing, a panel of 24 cultures, 8 non-proliferating and 16 proliferating, was selected and sequenced for CNAs and mutations and *CDKN2A* methylation status (Figure 4). In only 1 out of 24 cultures (LP123), there were no genetic alterations: neither copy number changes nor mutations in the sequenced genes. Next, we analyzed whether CNAs were associated with passage number and divided the group into cultures that stopped after initial plating (passage 0) or after one passage and cultures that were passaged more than once for differential analysis. There was no difference in the number of CNAs in the two groups (*p* = 0.823, Wilcoxon rank sum test). In addition, none of the specific genomic regions or genes were more often affected in passage >1 cultures compared to the other group (*p* > 0.05, Fisher’s exact test, Appendix A). 

The observed genetic changes appeared insufficient to sustain a replicative lifespan of OL cells that would allow for the screening for small molecule inhibitors or performing functional genetic screens. While the data suggest that there was no association between genetic changes and the number of population doublings, other factors may have influenced culture success, such as the quality and size of the biopsy.

In total, we were able to generate cell lines from three OL biopsies: VU-preSCC-LP084 (LP084), VU-preSCC-LP121 (LP121), and VU-preSCC-LP140 (LP140). An overview of the clinical background of the patients, mutations present in the original biopsies, and performed genomic engineering is provided in Table 1. Genomic engineering resulted in only one cell line with an extended lifespan (LP121-p16^KO^) and three immortalized cell lines (LP084-TERT^+^, LP121-p16^KO^-TERT^+^, and LP140-p16^KO^-TERT^+^, Figure 4). Again, *CDKN2A* KO or loss was present in all four cell lines (Figure 4, Appendix A), either natural or engineered. It should be noted that there was still p16 protein expression observed in LP140-p16^KO^-TERT^+^, although we did not assess the functionality of this protein (Appendix A). In addition, in all immortalized cell lines, ectopic TERT was introduced. *TP53* was only mutated in immortalized cell line LP084-TERT^+^. Interestingly, the double loss of *CDKN2A* and the mutation in *TP53* present in LP084-TERT^+^ were either acquired during cell culturing or cells carrying the mutation were selected upon *TERT* expression. Overall, genomic engineering was efficient in proliferating cells as, from all cell cultures that were manipulated, at least one or more immortalized cell lines were obtained. This was very effective for normal and tumor-adjacent oral keratinocytes as these cultures always proliferated for a number of passages, but effectivity in OL was very limited because genomic engineering was only possible for 3 out of 35 cultures. 

### 3.2. Genetic Aberrations Introduced in oral Keratinocytes by Genomic Modification and/or Prolonged Culturing

Many genetic changes became apparent after long-term cell culture, being undetectable at baseline (Figure 1, Figure 2 and Figure 3 and Appendix A). OL lesions could be heterogeneous, which would mask mutations and/or CNAs at baseline, and long-term proliferating cultures could have been the result of in vitro clonal selection. The detection limit of identifying such clones depended on the applied sequencing depth but was generally at least 0.1% for mutations and 5–10% for CNAs. Given this sensitivity, we assume that many genetic changes were induced in vitro. In line with this, we observed more CNAs in genetically modified cultures compared to the parental cultures as the former were sequenced after a significantly higher number of population doublings compared to the latter (Appendix A). Whether these additional CNAs were due to replication stress during prolonged culturing, oxidative stress, genetic manipulation with Cas9 expression in general, the specific KO of *CDKN2A* or *TP53*, or even the induction of *TERT* remains to be investigated. Increases in the number of mutations in OSCC driver mutations were seldom observed. 

To study this phenomenon in more detail, three cell lines were selected and samples were sequenced at different time points during prolonged proliferation (Appendix A: (A) LP140-p16^KO^-TERT^+^, (B) HN1037-p16^KO^-p53^KO^, and (C) HN1037-p16^KO^-TERT^+^). All cell lines started out with subclonal mutations in the targeted genes at baseline and, after prolonged culturing, only one or a few of those mutations persisted. In addition, one additional subclonal mutation was observed in *RB1* in LP140-p16^KO^-TERT^+^ at 64 population doublings (Appendix A). Subclonal CNAs were observed in all three lines in the first sequenced sample. While some of these were lost and others persisted, it was clear that in all three cell lines, additional clonal CNAs emerged at the later population doublings (Appendix A). The strong clonal outgrowth of cells with CNAs on chromosomes 5 and 7 between PD43 and PD64 for LP140-p16^KO^-TERT^+^ was highly suggestive of de novo CNAs. As the *CDKN2A* mutations that were introduced by genomic engineering were present at these population doublings in more than 99% of the cells, it is very likely that the observed subclonal CNAs were introduced after the modifications. 

### 3.3. Assessment of Drug Effectivity in the Oral Cell Culture Panel

The primary aim of this study was to generate a panel of OL and other precancer cell cultures suitable for drug testing. To find differences in drug efficacy due to divergent genetic or epigenetic makeup besides genomic engineering, we decided to test a panel of novel precancer cell lines with ectopic *TERT* expression and *CDKN2A* KO (natural or engineered): LP084-TERT^+^, LP121-p16^KO^-TERT^+^, LP140-p16^KO^-TERT^+^, M3-TERT^+^, HN1028-p16^KO^-TERT^+^, and UPPP60-p16^KO^-TERT^+^. In addition, two control cell lines were included for each drug: UM-SCC-22A as an HNSCC reference and UPPP60 as a normal reference cell line, which were used to provide a range of drug sensitivities. The effect on cell viability of the following drugs was assessed: GSK461364, LY2606368, MK1775, and S63845. S63845 was most effective in the modified cell lines compared to the control cell lines, and the data are therefore included in Figure 5. The data on the other three drugs are depicted in the Appendix A (GSK461364, Appendix A; LY2606368, Appendix A; MK1775, Appendix A). The targets of these four small molecule inhibitors have previously been identified in genome-wide siRNA screens, and both the inhibitors and their respective targets were further validated in the context of oral precancer and HNSCC [38,39,40,62,63]. GSK461364 is a PLK1 inhibitor, which is an essential cell cycle regulator previously shown to be a good target in HNSCC [39]. The inhibition of PLK1 was similarly effective across modified lines, but not as effective as in the sensitive tumor line. In contrast, inhibiting PLK1 did not seem to have any effect on cell viability in normal oral keratinocytes (Appendix A). These results indicate a relatively wide therapeutic window for PLK1 inhibition. LY2606368 is a Chk1 inhibitor, another important cell cycle regulator: the blocking of Chk1 results in the inhibition of cyclin-CDK complexes, resulting in halted cell proliferation [40]. All modified cell lines showed sensitivity to Chk1 inhibition several orders of magnitude below the IC50 of the normal keratinocytes. It should be noted that the effects of the inhibitor plateaued in several cell lines, suggesting that other proteins may take over the function of Chk1 or that off-target effects impact kinases in proliferation or cell death (Appendix A). MK1775 is a dual inhibitor of PLK1 and Wee1, another protein involved in the formation of cyclin-CDK complexes. Wee1 inhibition was also effective in all modified cell lines, with all IC50s between the reference values, and 10-100x more effective compared to the insensitive normal oral keratinocytes. Again, a plateau was visible in two lines (LP121-p16^KO^-TERT^+^ and LP140-p16^KO^-TERT^+^), but this effect was less pronounced compared to the inhibition of Chk1 (Appendix A). In addition to targeting cell cycle regulators, S63845 was used to inhibit MCL1, an anti-apoptotic BCL family member. The inhibition of MCL1 promotes apoptosis, eventually leading to cell death [64]. MCL1 inhibition showed only a small therapeutic window between the two reference cell lines and the effectivity in the different cell lines was more varied compared to the other inhibitors. LP140-p16^KO^-TERT^+^ and HN1028-p16^KO^-TERT^+^ were resistant to MCL1 inhibition, while M3-TERT and UPPP60-p16^KO^-TERT^+^ had similar IC50 values to the sensitive UM-SCC-22A cell line. LP121-p16^KO^-TERT^+^ and M3 were somewhat more sensitive, while LP084-TERT^+^ was several factors more sensitive to MCL1 inhibition compared to the sensitive reference (Figure 5).

## 4. Discussion

Improving the treatment of OL has been hampered by a lack of suitable model systems that can be used for the identification of druggable targets and the testing of drug efficacy. Although OL cell models have been established in the past, these are rare and selected. We therefore aimed to generate a broad panel of new OL cell lines using either spontaneous growth in vitro or by CRISPR/Cas9 genomic engineering. We first showed that by the right combinations of genomic modifications, it is possible to create immortalized cell lines from normal and tumor-adjacent keratinocytes with very high efficiency. However, the translation of this approach to OL was hampered by the relatively poor success rate of short-term cultures and the associated limited time frame available for in vitro experimentation of OL cultures. In our experimental setup, we only performed genomic modification when cultures reached sufficient confluency for passaging. For a higher yield of extended OL cultures, immediate *CDKN2A* knockout and ectopic *TERT* expression were considered. 

*CDKN2A* clearly played a key role as all prolonged proliferating and immortalized lines contained mutations or losses of this gene. This corroborates the results found by other studies in which immortalized cell lines were generated from oral keratinocytes [44,45,46,47]. In this study, the loss of encoded p16 was found to be particularly crucial in order to prevent premature senescence in primary oral keratinocytes, which may have been due to direct contact with plasticware [65]. 

Several of our immortalized cultures were derived without a naturally inactivated or knocked-out *TP53*. This is in contrast with models in which p16-insensitive CDK4 was used, suggesting that the role of *TP53* in preventing immortalization may be neutralized independently of CDK4, for instance, through the loss of ARF [43,66]. Sole KO of *TP53* on the other hand did not effectively result in immortalization or even extended lifespans, as was also reported for dominant-negative mutations in *TP53* [43]. The role of dominant-negative mutations versus the loss of p53 function is intriguing in these models since roughly half of the mutations in *TP53* are missense [36,67]. 

Also, *TERT* activation plays a key role in immortalization of oral keratinocytes. Activating *TERT* promotor mutations are frequently found in HNSCC, especially in OSCC, which is the reason that we included the *TERT* promotor in our gene panel for target enrichment sequencing [68,69]. However, we did not observe a *TERT* promotor mutation in any of our cell cultures. Presumably, these constitute a later step in carcinogenesis as, in tumor samples, *TERT* promotor mutations appear in as much as half of all OSCC (Muijlwijk et al., manuscript in preparation) [69]. In two cases, the activation of *TERT* appeared to be sufficient for immortalization but, in both models, the cells already contained a natural double loss of *CDKN2A* or lost *CDKN2A* during the process. Inversely, exogenous *TERT* activation was required for immortalization in *CDKN2A* KO lines unless *TP53* was also lost. A clear relationship has been demonstrated between *TP53* alterations and telomere lengthening in several tumor types [70]. Telomeres may be lengthened independently from telomerase in a homologous recombination-based process known as the alternative lengthening of telomeres, which is especially associated with mutated *TP53* [71,72,73]. Therefore, to overcome senescence, it may be possible that oral keratinocytes, in addition to mutated *CDKN2A*, require the maintenance of telomeres, either canonically through the activation of *TERT* or through the alternative lengthening of telomeres facilitated by mutations of *TP53*. 

The protocol for the generation of short-term cell cultures (up to 20 population doublings) was previously applied for normal oral keratinocytes and tumor-adjacent mucosal cells and had a success rate of over 90% [47]. As stated above, for OL biopsies, the success rate was very low as keratinocytes only emerged from one in three OL biopsies and, even when proliferation was observed, only a very limited number of samples was growing sufficiently to enable genomic engineering to increase the lifespan. This difference may be attributed to the size of the biopsy as OL biopsies tend to be rather small, especially when the biopsy is only incisional. Moreover, the changed properties of the cells may have made it difficult to separate the layers and keep the basal cell layers attached to the superficial layers. We anticipated this potential problem and always cultured the remaining tissue as well, but that did not improve the results. Due to the nature of the tissue, the viability of OL cells may be somewhat lower. In a study in which cell lines were generated from primary patient-derived biopsies from a wide range of cancer types, it was shown that the success rate of culturing is highly dependent on the number of viable cancer cells present in the biopsy [74]. 

A different approach to generating oral keratinocyte cell lines is through the use of feeder layers. A ~40% success rate was reported for the generation of immortalized cell lines on a feeder layer consisting of irradiated fibroblasts, with all included biopsies growing at least for some population doublings [44,45,46]. This feeder layer secretes a collagen matrix that prevents contact between the oral keratinocytes and the plastic surface, which normally induces p16-mediated senescence [65,75]. While the use of a feeder layer may enable more efficient generation of cell lines from oral biopsies, it may interfere with drug sensitivity assays and genetic analyses [74,76]. Removal of the feeder layer after initial culturing drastically reduces cell proliferation rates and, in the case of HNSCC, more than 90% of cultures stop growing [44,74]. More recently, the feeder layer method was optimized and now includes the use of the Rho kinase inhibitor, which further delays senescence and enables the efficient use of CRISPR-Cas9 [77,78]. The question remains whether these cultures still resemble OL after these modifications. These questions may be answered in further investigations of the lines in terms of transcriptomic and metabolic activity, as well as the functional characterization of invasive properties and tumorigenic potential. 

More recently, HNSCC cells have been grown in Matrigel (or equivalent matrices), sometimes in co-cultures with fibroblasts, because 3D-cultured organoids might better mimic the in vivo characteristics of the disease [79,80,81]. With this method, it was possible to generate multiple HNSCC cell lines without the use of feeder layers or other modifications [79]. This process was optimized by the addition of cancer-associated fibroblasts [81]. While only culture lifespan was extended and cell lines were not immortalized, these organoids proliferated long enough for drug assessment assays [79]. It remains to be seen whether this approach might work for OL as well. 

Prolonged proliferation in vitro resulted in the clonal selection of CNAs and mutations, and genetic changes may have been induced. It is difficult to formally exclude that these changes were present at baseline in a single or very small number of cells just below the sequencing coverage threshold, but, given the sequencing depth, this seems unlikely. Our data are more suggestive of genomic instability introduced by the KO of *TP53* and *CDKN2A* [82] and the associated prolonged culturing. The loss of *TP53* and *CDKN2A* caused replication stress in cell cultures and mouse models, inducing DNA damage [83,84]. Also, the role of Cas9 itself may impact genomic stability, as CRISPR-mediated genome editing may introduce gross chromosomal rearrangements and CNAs [85,86,87]. However, cultures that immortalized without any genomic modification with CRISPR/Cas9 (LP084 and M3) displayed a variety of genetic changes during prolonged propagation, again suggesting that replication stress during prolonged proliferation may be the root cause of genomic instability. The high proliferation rates and high oxygen pressure, combined with the loss of *CDKN2A* and *TP53*, likely cause genomic changes over time. The acquisition of culturing-induced genetic variability constitutes a limitation and potential pitfall of in vitro precancer models and should be taken into account for future research, including when organoid-like models are employed. 

The four investigated drugs proved to be effective in most of the generated oral precancer cell lines. PLK1 inhibitor GSK461364 was effective in all cell lines developed, while PLK1 inhibition did not result in any cell killing in the normal oral keratinocytes. These results are comparable to those of previous studies investigating the use of GSK461364 for the treatment of HNSCC cell lines [39]. However, GSK461364 was delivered intravenously and several side effects were reported, making it somewhat less suitable for OL treatment [41]. As PLK1 inhibition provides a substantial therapeutic window, it may be interesting to investigate novel PLK inhibitors such as onvansertib, which is administered orally and shows promising results in the treatment of HNSCC [88]. Chk1 inhibitor LY2606368 showed a substantial decrease in cell viability in modified oral cell lines compared to the insensitive normal oral keratinocytes, which is similar to studies in which the effects of Chk1 inhibition in HNSCC were investigated [40]. Again, a phase one trial in which LY2606368 was assessed for the treatment of HNSCC reported a wide range of side effects, but this was in combination with chemoradiotherapy [89]. Wee1 inhibition was also comparably effective as previously reported [38]. In clinical trials, Wee1 inhibition by MK1775 was associated with side effects [90]. It has been shown that MK1775 is actually a dual inhibitor targeting both Wee1 and PLK1 [91]. It might be better to investigate the use of a more selective drug that only targets Wee1, such as Debio-0123 or Zn-C3, which are currently in clinical trials for the treatment of solid tumors [92,93]. MCL1 inhibitor S63845 was the only drug that was more effective in several of the modified lines compared to the sensitive tumor line. The inhibition of anti-apoptotic factors has only recently emerged as a viable treatment pathway for HNSCC, and seems to be effective in vitro, but in vivo side effects are currently unknown [94]. The remarkable sensitivity of one of the precancer lines proves that expanding precancer model systems may reveal drug sensitivities, hinting that personalized treatment might be possible. This is of particular interest in the precancer setting, which permits a longer time frame to find a personalized treatment compared to OSCC. Overall, the investigated drugs show efficacy in the generated cell lines, but the results should be interpreted with caution.

## 5. Conclusions

The methods described in this study yielded various immortalized tumor-adjacent and OL cell lines. As in 2D models, it was difficult to establish OL cell lines, 3D-organoid-like cultures may be the best way forward as they more closely resemble the in vivo situation, without the need for any modifications that may influence drug testing experiments. However, as genomic instability has already been observed in other cancer organoids, close monitoring of the genetic stability of these models is also warranted [95,96].

## Figures and Tables

**Figure 1 cells-13-00710-f001:**
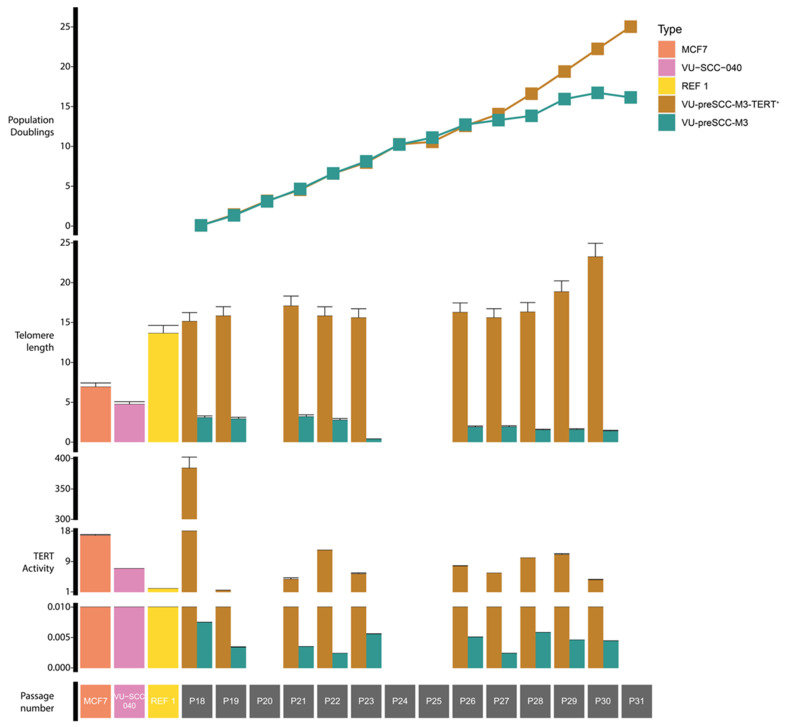
Changes in telomere length and telomerase activity over time between VU-preSCC-M3 and VU-preSCC-M3-TERT^+^. The top graph shows the population doublings for both cell lines over time. The middle graph shows the changes in average telomere length over time per chromosome end. The bottom graph shows the changes in telomerase activity over time. In addition, three control samples were included with known telomere lengths and telomerase activities: MCF-7, VU-SCC-040, and the reference sample from the used kits. The bottom row shows the control samples and the cell passages for the M3 cell cultures.

**Figure 2 cells-13-00710-f002:**
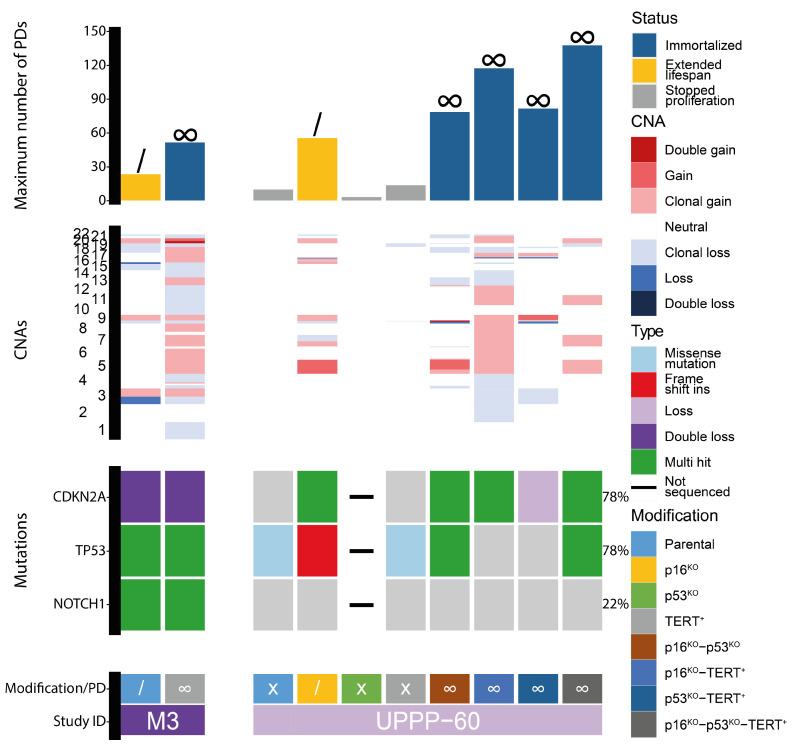
Overview of population doublings and genetic aberrations in a tumor-adjacent and normal oral keratinocyte cell culture. The top graph shows the number of population doublings for each included culture. The color of each bar indicates the proliferation status for each culture. In addition, a “/” indicates extended lifespan while “∞” indicates that the cell line was immortal. In the middle graph, the CNAs for each culture are presented, with the chromosomes on the y-axis. The bottom graph contains the mutations present in each culture. The colors indicate the types of mutations. Below the graphs, the modifications for each culture are indicated and the proliferation status is provided, where “X” indicates limited proliferation, “/” indicates extended lifespan, and “∞” indicates immortalization. In addition, the study ID for each set of cultures is provided. PDs = population doublings. CNA = copy number aberration.

**Figure 3 cells-13-00710-f003:**
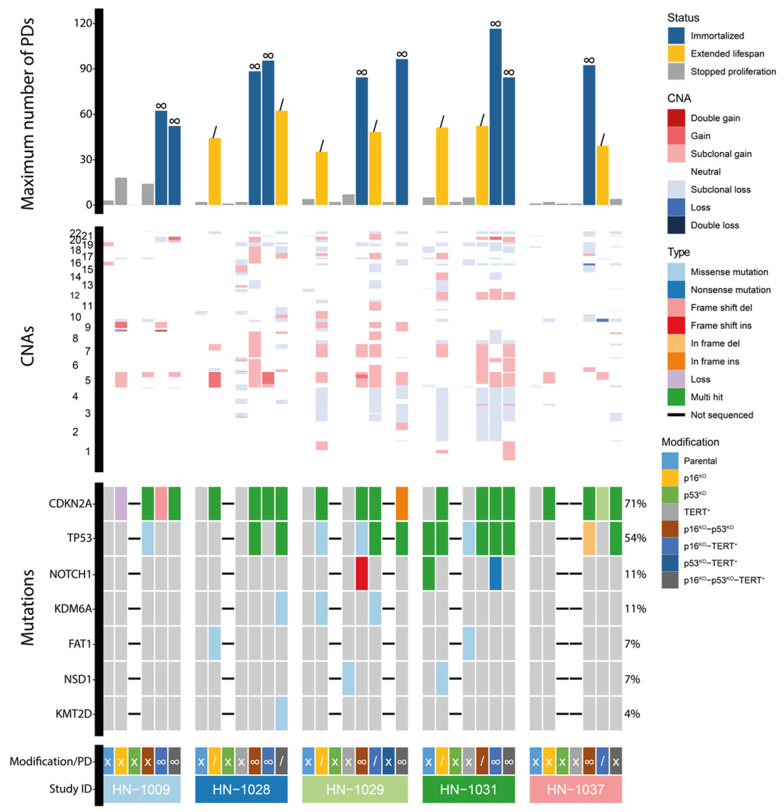
Overview of population doublings and genetic aberrations in five tumor-adjacent oral keratinocyte cell cultures. The top graph shows the number of population doublings for each included culture. The color of each bar indicates the proliferation status for each culture. In addition, a “/” indicates extended lifespan while “∞” indicates that the cell line was immortal. In the middle graph, the CNAs for each culture are presented, with the chromosomes on the y-axis. The bottom graph contains the mutations present in each culture. The colors indicate the types of mutations. Below the graphs, the modifications for each culture are indicated and the proliferation status is provided, where “X” indicates limited proliferation, “/” indicates extended lifespan, and “∞” indicates immortalization. In addition, the study ID for each set of cultures is provided. PDs = population doublings. CNA = copy number aberration. del = deletion. ins = insertion.

**Figure 4 cells-13-00710-f004:**
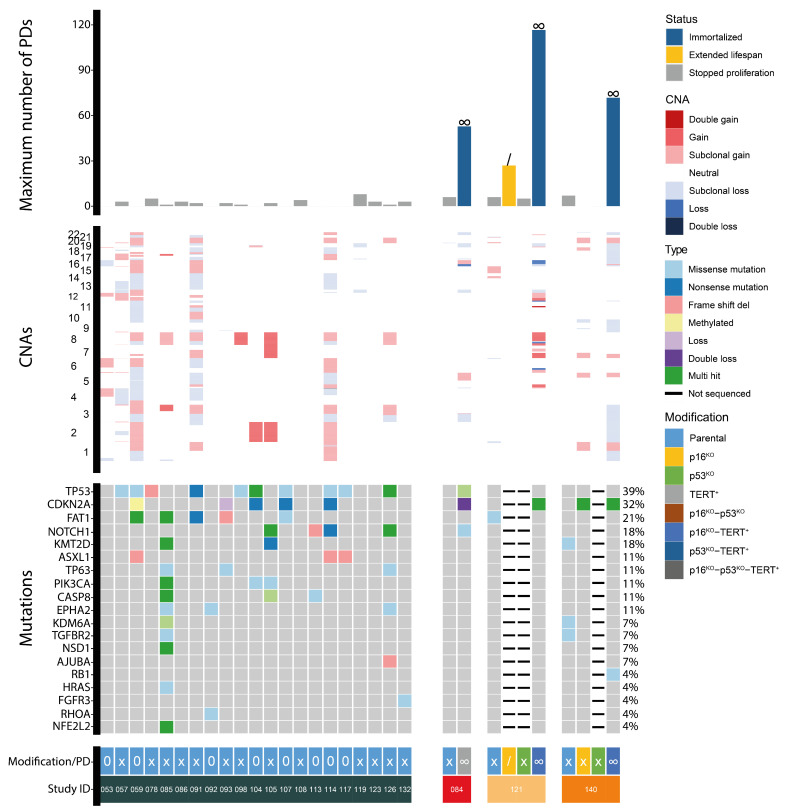
Overview of population doublings and genetic aberrations in a panel of oral leukoplakia cultures, including three that were genetically modified. The top graph shows the number of population doublings for each included culture. The color of each bar indicates the proliferation status for each culture. In addition, a “/” indicates extended lifespan while “∞” indicates that the cell line was immortal. In the middle graph, the CNAs for each culture are presented, with the chromosomes on the y-axis. The bottom graph contains the mutations present in each culture. The colors indicate the types of mutations. Below the graphs, the modifications for each culture are indicated and the proliferation status is provided, where “0” indicates no proliferation, “X” indicates limited proliferation, “/” indicates extended lifespan, and “∞” indicates immortalization. In addition, the study ID for each set of cultures is provided. PDs = population doublings. CNA = copy number aberration. del = deletion.

**Figure 5 cells-13-00710-f005:**
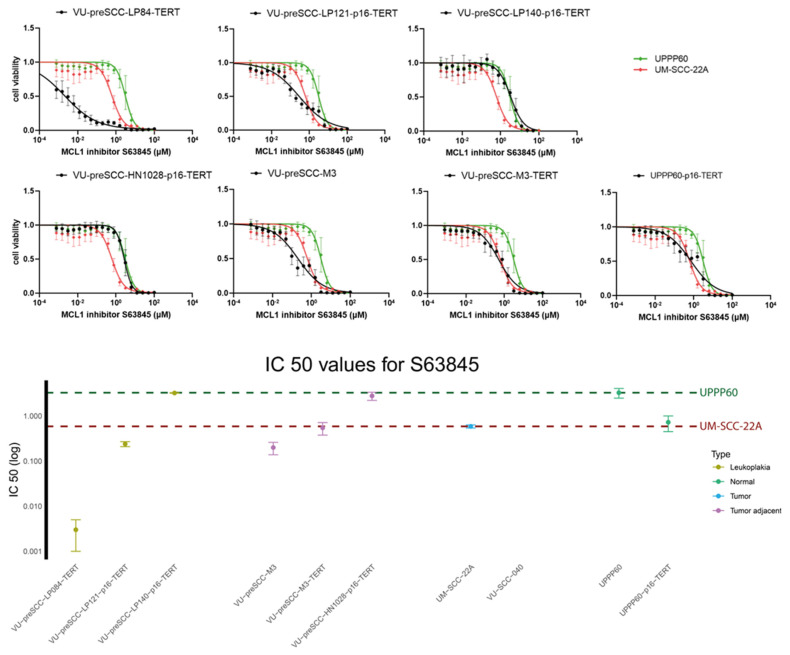
Effectivity of MCL1 inhibitor S63845 for OL treatment. (**Top**): Dose–response curves showing the relative cell viability of modified cell lines (black) with sensitive tumor line UM-SCC-22A (red) and epithelial line UPPP60 (green) as a reference indicating the therapeutic window of MCL1 inhibitor S63845. Experiments were performed 3 times in triplicate and the averaged value of the 3 experiments is presented. (**Bottom**): Plot showing the IC50 for MCL1 inhibitor S63845 in all included cell lines. Samples are sorted based on cell type as indicated by color. The dotted red line indicates the IC50 for UM-SCC-22A, defined as the sensitive cell line. The dotted green line indicates the IC50 for UPPP60, defined as the insensitive cell line.

**Table 1 cells-13-00710-t001:** Overview of clinical variables for patients from which biopsies were obtained and put in cultures and that were subsequently modified to improve growth. Modifications were either singular or performed in combination, as indicated by the + sign. KO indicates that a knockout was performed for the specified gene, while + indicates that the gene was ectopically expressed.

Patient ID	Gender	Age at Biopsy	Location	TNM	Mutations Present in Parental Culture	Modifications
LP084	Female	74	Floor of mouth	-	None	TERT^+^
LP121	Male	51	Tongue	-	*FAT1*	p53^KO^, p16^KO^, p16^KO^ + TERT^+^
LP140	Male	62	Tongue	-	*KMT2D*, *TGFBR2*, *KDM6A*	p53^KO^, p16^KO^, p16^KO^ + TERT^+^
M3	Male	67	Tumor-adjacent tissue of a larynx carcinoma	pT4aN0 (TNM7)	*TP53*, *NOTCH1*, focal 9p double loss	TERT^+^
HN1009	Male	65	Tumor-adjacent tissue of a hypopharynx carcinoma	pT4aN3b	None	p53^KO^, p16^KO^, p16^KO^ + p53^KO^, p16^KO^ + TERT^+^, p16^KO^ + p53^KO^ + TERT^+^
HN1028	Male	85	Tumor-adjacent tissue of a larynx carcinoma	pT4aN0	None	p53^KO^, p16^KO^, TERT^+^, p16^KO^ + p53^KO^, p16^KO^ + TERT^+^, p16^KO^ + p53^KO^ + TERT^+^
HN1029	Male	67	Tumor-adjacent tissue of a larynx carcinoma	pT4aN1	None	p53^KO^, p16^KO^, TERT^+^, p16^KO^ + p53^KO^, p53^KO^ + TERT^+^, p16^KO^ + TERT^+^, p16^KO^ + p53^KO^ + TERT^+^
HN1031	Male	58	Tumor-adjacent tissue of a hypopharynx carcinoma	pT4aN3b	*TP53*, *NOTCH1*	p53^KO^, p16, TERT^+^, p16^KO^ + p53^KO^, p16^KO^ + TERT^+^, p16^KO^ + p53^KO^ + TERT^+^
HN1037	Female	57	Tumor-adjacent tissue of a lip and oral cavity carcinoma	pT3N2b	None	p53^KO^, p16^KO^, TERT^+^, p16^KO^ + p53^KO^, p16^KO^ + TERT^+^, p16^KO^ + p53^KO^ + TERT^+^
UPPP-60	Male	Unknown	Uvula	-	*TP53*	p53^KO^, p16^KO^, TERT^+^, p16^KO^ + p53^KO^, p53^KO^ + TERT^+^, p16^KO^ + TERT^+^, p16^KO^ + p53^KO^ + TERT^+^

## Data Availability

Sequencing data or cell lines may be obtained after a formal request and after an MTA has been signed.

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
