# Peer review of "Genomic Engineering of Oral Keratinocytes to Establish In Vitro Oral Potentially Malignant Disease Models as a Platform for Treatment Investigation"

_cells, 2024, doi:10.3390/cells13080710_

Round 1

Reviewer 1 Report

Comments and Suggestions for Authors

The manuscript by Wils et al. established several cell culture models to be used in drug screening in oral cancer.  The reviewer is amazed by the work performed in this study, and the meticulous analysis of the cell cultures and the pharmacological testing.

Minor comments:

Dispase was for 4 hr at RT+ 15 min at 37 ? please re-check (line 150-3)

Please confirm that the cells were not grown on coated plates? 

the selection of the inhibitors and the division between supplemented figures and the main text is not obvious and explanation to the readers is in order

replace lines  616 -618 "This section may be...." to a short summary of that part 

Author Response

Dear Sir/Madam,

We thank Reviewer 1 for the time spent reading our manuscript and providing relevant comments. Below we have provided a point to point answer for each individual comment.

Reviewer #1:

The manuscript by Wils et al. established several cell culture models to be used in drug screening in oral cancer.  The reviewer is amazed by the work performed in this study, and the meticulous analysis of the cell cultures and the pharmacological testing.

Minor comments:

Dispase was for 4 hr at RT+ 15 min at 37 ? please re-check (line 150-3)

This is correct. The 4 hours RT is used for dispase diffusion in the tissue and the 15 minutes at 37°C is used for activation of the dispase. For clarification, we have modified the following sentences in the Materials and Methods, Processing of oral leukoplakia and tumor adjacent biopsies section, page 4, lines 149-154:

“Surgically removed fresh OL and tumor-adjacent biopsies were cut into smaller pieces (depending on biopsy size) and transferred to a 1X Dispase II (cat. no. 4942078001, Roche, Basel, Switzerland) and phosphate-buffered saline (PBS, cat. no. BE17-517Q, Lonza, Verviers, Belgium) solution and incubated at room temperature (RT) for 4 hours to allow diffusion. Subsequently, the vial was incubated at 37°C for 15 minutes to activate the dispase and to allow separation of the mucosal epithelium and submucosal layers.”

Please confirm that the cells were not grown on coated plates?

               The cells were not grown on coated plates but on standard TC plates.

the selection of the inhibitors and the division between supplemented figures and the main text is not obvious and explanation to the readers is in order

For the sake of brevity, we wanted to include the data of one drug as a Figure in the main text and selected the drug that was most effective in the modified cell lines compared to the control cell lines, while the Figures of the other drugs are presented in the Supplementary data. To make this more clear the following sentences were modified in the Results, Assessment of drug effectivity in the oral cell culture panel section, page 13, lines 544-549:

“The effect on cell viability of the following drugs was assessed: GSK461364, LY2606368, MK1775 and S63845. S63845 was most effective in the modified cell lines compared to the control cell lines and the data were therefore included in Figure 5. The data on the other three drugs are depicted in Supplementary figures (GSK461364, Supplementary Figure S8; LY2606368, Supplementary Figure S9; MK1775, Supplementary Figure S10).”

replace lines  616 -618 "This section may be...." to a short summary of that part

These lines were erroneously placed in the main body of the text. The lines have now been removed.

Reviewer 2 Report

Comments and Suggestions for Authors

This study was developed with two main objectives: 1) to generate cultures from oral leukoplakia (OL) biopsies resulting in oral precancer cell lines with an extended lifespan, that may subsequently be used for investigation of novel preventive treatments; 2) to characterize the genetic factors involved. For this purpose, the authors performed several experiments to show that addition of the TERT gene produces immortalization of oral squamous cell carcinoma tumor-adjacent cell culture. Subsequently, they demonstrate that it is possible to generate immortalized cell lines from normal oral keratinocytes and tumor-adjacent oral biopsies using genomic engineering to modify selected genes. This same approach is applied to generate immortalized OL cell lines, and these cultures are genetically characterized. Finally, a subset of immortalized cell lines is used to assess the response of a set of targeted molecule inhibitors. Despite the interesting findings of the work, some points need to be better addressed:

 Major points:

 - Material and Methods section – Dose response analysis: The authors cited four inhibitors used: Nutlin-3a, MK1775, S63845, or LY2606368. However, in the Results section they described the effects of GSK461364 (S7), LY2606368 (S8), MK1775 (S9) and S63845 (Figure 5) inhibitors. Check.

 -          Include in the Material and Methods section – Cell culturing – the description of VU-UPPP60 cell lineage.

-          Provide western blot analysis to show P53 knockdown, as demonstrated for P16.

 -  In the tests carried out with the four inhibitors (GSK461364, LY2606368, MK1775 and S63845), how do the authors ensure that the results obtained are in fact related to the inhibition of the different protein targets?

 Minor points:

 -          Include Supplementary figure legends

Author Response

Dear Sir/Madam,

We thank Reviewer 2 for the time spent reading our manuscript and providing relevant comments. Below we have provided a point to point answer for each individual comment.

Reviewer #2:

This study was developed with two main objectives: 1) to generate cultures from oral leukoplakia (OL) biopsies resulting in oral precancer cell lines with an extended lifespan, that may subsequently be used for investigation of novel preventive treatments; 2) to characterize the genetic factors involved. For this purpose, the authors performed several experiments to show that addition of the TERT gene produces immortalization of oral squamous cell carcinoma tumor-adjacent cell culture. Subsequently, they demonstrate that it is possible to generate immortalized cell lines from normal oral keratinocytes and tumor-adjacent oral biopsies using genomic engineering to modify selected genes. This same approach is applied to generate immortalized OL cell lines, and these cultures are genetically characterized. Finally, a subset of immortalized cell lines is used to assess the response of a set of targeted molecule inhibitors. Despite the interesting findings of the work, some points need to be better addressed:

 Major points:

 - Material and Methods section – Dose response analysis: The authors cited four inhibitors used: Nutlin-3a, MK1775, S63845, or LY2606368. However, in the Results section they described the effects of GSK461364 (S7), LY2606368 (S8), MK1775 (S9) and S63845 (Figure 5) inhibitors. Check.

Nutlin-3a was erroneously mentioned and therefore removed from the text. For additional clarification, The following sentences were modified in the Materials and Methods, Dose-response analysis section, page 9, lines 363-370:

“Subsequently, the efficacy of four different small molecule inhibitors was assessed by serial dilutions. Serial dilutions for Wee1 inhibitor MK1775 (cat. no. 2373, Biovision, San Francisco, CA, USA), MCL1 inhibitor S63845 (cat. no. HY-100741, MedChemExpress, Monmouth Junction, NJ, USA) or Chk1 inhibitor LY2606368 (cat. no. HY-18174, MedChemExpress, Monmouth Junction, NJ, USA) ranged from 0.8 nM to 100 µM. For PLK1 inhibitor GSK461364 (cat. no. Axon 1688, Axon Medchem, Reston, VA, USA) a serial dilution ranging from 7.6 pM to 1 µM was used.”

 -          Include in the Material and Methods section – Cell culturing – the description of VU-UPPP60 cell lineage.

We have included the origin and cited references of all previously established cell lines in the Materials and Methods, Cell culturing section, pages 4-5, lines 166-174:

“Precancer cell line VU-preSCC-M3 (M3)[48], tumor cell line VU-SCC-040 (040)[49] and normal oral keratinocyte cell line VU-UPPP60[47] were previously generated at our own department. Tumor cell lines UM-SCC-11B (11B)[50], UM-SCC-17A (17A)[50], UM-SCC-22A (22A)[50] and UM-SCC-47 (47)[50] were obtained from Prof T. Carey (University of Michigan, Ann Arbor, MI, USA). Human breast cancer cell line MCF-7[51], used as a control in several experiments, and the HEK293T cell line, used for lentiviral packaging, were both acquired from ATCC (MCF7: HTB-22; HEK293T: 293tsA1609neo; ATCC, Manasses, VI, USA).”

-          Provide western blot analysis to show P53 knockdown, as demonstrated for P16.

We have performed an additional western blot to show TP53 knockdown, which is included as Supplementary Figure S5. In addition the following sentences have been modified:

Material and Methods: Western Blot section, page 9, lines 349-354: “Western blots were performed using p16 mouse monoclonal antibody (p16 INK4A (JC8), sc-56330, cat. no. C1121, Santa Cruz Biotechnology, Dallas, TX, USA) and p53 mouse monoclonal antibody (p53 Protein (DO-7), cat. no. M7001, Agilent, Santa Clara, CA, USA) and visualized with red fluorescent label goat anti-mouse secondary antibody (LI-COR IRDye 680 RD, 926-68070, cat. no. D11130-05, LI-COR Biosciences, Lincoln, NE, USA), or ECL.”

Results: Immortalization of oral keratinocytes by genomic engineering of selected target genes section, page 11, lines 425-428: “KOs were confirmed by NGS and analysis of depth of coverage and IGV inspection when necessary, and loss of protein expression was confirmed by Western blot (Figure 2, Supplementary Figures S1-S5).”

 -  In the tests carried out with the four inhibitors (GSK461364, LY2606368, MK1775 and S63845), how do the authors ensure that the results obtained are in fact related to the inhibition of the different protein targets?

All four small-molecule inhibitors were included based on previous research performed at our department where these targets were identified in siRNA screens and subsequently validated using the same inhibitors. We can therefore be sure that these inhibitors affect cell viability through their respective targets. The behavior of siRNAs in serial dilution and the drugs in serial dilutions was very identical with respect to efficacy and therapeutic window (De Boer et al. Oncotarget 2017). This does not imply that the inhibitors are completely free of additional off-target effects, but it does indicate that the cell death phenotype can be well explained by direct inhibition of that particular gene product.

In addition the following sentence was added to the Results, Assessment of drug effectivity in the oral cell culture panel section, page 14, lines 550-553: “The targets of these four small-molecule inhibitors have previously been identified in genome-wide siRNA screens and both the inhibitors and their respective targets were further validated in the context of oral precancer and HNSCC[38–40,62,63].”

 Minor points:

 -          Include Supplementary figure legends

               We have added Supplementary figure legends.

Round 2

Reviewer 2 Report

Comments and Suggestions for Authors

In the revised submission the authors have satisfactorily addressed my comments and made the necessary changes to the manuscript.